# Peak-Load Forecasting for Small Industries: A Machine Learning Approach

**Dong-Hoon Kim**, **Eun-Kyu Lee \*** and **Naik Bakht Sania Qureshi**

Department of Information and Telecommunication Engineering, Incheon Nat'l University, Incheon 22012, Korea; dhkim85@inu.ac.kr (D.-H.K.); saniaq2017@inu.ac.kr (N.B.S.Q.)
**\*** Correspondence: eklee@inu.ac.kr; Tel.: +82-32-835-8629

**Abstract:** Peak-load forecasting prevents energy waste and helps with environmental issues by establishing plans for the use of renewable energy. For that reason, the subject is still actively studied. Most of these studies are focused on improving predictive performance by using varying feature information, but most small industrial facilities cannot provide such information because of a lack of infrastructure. Therefore, we introduce a series of studies to implement a generalized prediction model that is applicable to these small industrial facilities. On the basis of the pattern of load information of most industrial facilities, new features were selected, and a generalized model was developed through the aggregation of ensemble models. In addition, a new method is proposed to improve prediction performance by providing additional compensation to the prediction results by reflecting the fewest opinions among the prediction results of each model. Actual data from two small industrial facilities were applied to our process, and the results proved the effectiveness of our proposed method.

**Keywords:** ensemble; isolation forest; machine learning; peak-load forecasting; small industry

---

## 1. Introduction

Peak-load forecasting prevents the waste of energy and with helps environmental issues by establishing plans for the use of renewable energy. In other words, peak-load prediction is necessary to use as part of simulation data for energy policy establishment. From an operator's point of view, by supplying stable power to the system, it is possible to operate facilities without any problems, such as power outages, and reduce costs. In addition, by comparing predicted results with actual measured values, system security evaluation can be used to address various vulnerabilities in advance. Therefore, various studies are continuously conducted for reliable peak-load prediction. Most of these studies focus on the expansion of existing prediction models or the development of new models, eliminating variables that are not conducive to prediction by selecting or extracting new feature information to increase accuracy. Thus, good predictive results can be obtained if the model proposed in these research results is trained by using appropriate data. Recently, various combinations of variables that can be used for peak-load prediction have been studied, and learning methods are evolving, so it is expected that the accuracy of peak-load prediction will gradually increased.

Peak-load forecasting is largely divided into two approaches: One is a time-series approach by using system load information over time, and the other is an AI-based approach, such as machine learning and deep learning, which creates a predictive model on the basis of information about various factors, such as output volume, weather information, number of workers, and their activities or behaviors that affect system load. Most time-series models assume that time-series data are stationary (statistical mean, standard deviation, and covariance are constant over time). The augmented Dickey–Fuller (ADF) or Kwiatkowski–Phillips–Schmidt–Shin (KPSS) test can be used to verify whether

data are stationary or not. For nonstationary data, time-series analysis can be performed after transforming them into stationary data through differencing, seasonal differencing, and transformation. However, if the assumptions of the model are incorrect, or the order of the mathematical model is not properly selected, prediction results are very poor. Daily peak-load prediction using time-series analysis generally aims to predict the peak load of the next day on the basis of system load information of the past few days. When forecasting with only information on a single variable, most use past peak-load information. When designing a model using multivariable information, on the other hand, meteorological information, such as temperature, humidity, and precipitation, is mainly used in addition to peak-load information. However, as the amount of feature information increases, modeling becomes complicated, and it is difficult to apply the prediction model implemented on the basis of a predefined mathematical assumption to various environments. Because of the advantage of being able to build an appropriate model without assuming a mathematical model, AI techniques are used rather than time-series analysis. Load prediction methods based on AI techniques require various input data types when predicting. Building automation and energy management systems provides reliable peak-load prediction by offering a variety of information to the model.

However, it is difficult for most small-scale industrial facilities to provide all input variables desired by this prediction model. This is because the infrastructure (e.g., advanced energy management systems) for collecting information is not established. Therefore, prediction performance is significantly lowered or fails to predict, as peak-load prediction is performed only with irregular load data that cannot only be explained by that information.

To overcome this problem, we analyzed two-year load data from two small industrial facilities in Silicon Valley, CA, USA. The acquired load data only provided power consumption information over time recorded over two years (2017–2018). These data contained very irregular daily peak-load information due to the inconsistent production of the facilities. The data of each facility were all nonstationary and had different statistical characteristics. Predicting the peak load of the next day by using historical peak-load information resampled from these data is very inaccurate, so we predicted the peak load of the same day on the basis of the load pattern of the day. In this case, the most important issue is when to predict the peak. This is because the prediction accuracy of the peak load that is too early is inevitably lowered, and if it is too late, the predicted information cannot be used. Therefore, we performed data analysis for a proper trade-off between prediction performance and prediction time point; then, we selected the appropriate feature information.

First, to explore the limitations of peak-load forecasting for typical small-scale industrial facilities, we analyzed predictive performance with respect to two small-scale industrial facility datasets by using typical peak-prediction models, such as bagging, AdaBoost, random forests, extra trees, and gradient tree boosting. In the case of Facility 1, where there were relatively no trends and seasonality, the $r^2$ value was close to 0, whereas in Facility 2, where trend and seasonality existed, an $r^2$ value of $-1.8$ was found. Although both facilities used commonly used predictive models, they showed very low predictability, especially for datasets including trends and seasonality.

To solve this, we introduce three ideas: First, a new feature selection method is proposed to predict daily peak power by grasping the load pattern for a short period of time from the time in which the equipment starts to operate, although the forecasting point is shorter than that of general short-term load prediction. Second, we propose a method that can generally cope with load patterns of various industrial facilities with different characteristics through an aggregated model of several ensemble models. The goal of ensemble methods is to combine the predictions of several base estimators built with a given learning algorithm in order to improve generalizability and robustness over a single estimator. The ensemble of models reduces model interpretability because of increased complexity, and makes it very difficult to draw any crucial business insights at the end; however, it had better performance on the test case scenarios (unseen data) as compared to individual models in most cases. Third, we propose to improve prediction performance by modifying the isolation forest. The isolation forest was originally introduced as a method for distinguishing between abnormal and normal samples

in a dataset, but we used this method to detect predictive outliers from the results of several predictive models. A prediction result that is determined as an outlier can be improved by providing additional compensation by using error distribution obtained during the training process.

As a result of implementing and experimenting with these ideas, we were able to lastly increase the $r^2$ scores of the facilities to 0.73 and 0.56, respectively. In addition, an effect of reducing the number of incidentally underestimated peak loads was obtained. Even with high forecasting accuracy, any underestimation of the peak load may result in power shortages in the spinning reserve of the system that may lead to it being insecure and unreliable. Through a series of processes that we propose, the number of underestimations was decreased from 120 to 48 and from 80 to 50 for Facilities 1 and 2, respectively.

The contributions of this paper are as follows.

- We propose a peak-load prediction model for small industrial facilities that provide only hourly power consumption information.
- We propose a new feature selection method that makes it possible to predict daily peak power by finding the load pattern for a short period of time from the time the facility is started.
- We propose an ensemble model, so that the peak-load-forecasting method for small industrial facilities could be extended in general cases.
- We modified the isolation-forest method to improve prediction accuracy.
- We suggest methods for evaluating underestimation.

Section 2 introduces various related studies for peak-load forecast in general industrial, commercial, and residential facilities. In Section 3, we apply well-known peak-load forecasting techniques to daily peak data of two actual small-scale industrial facilities, and analyze the limitations of these techniques. As a result of the experiment, the $r^2$ score value was close to 0 for one industrial facility without trends, and the other facility with trends had a negative $r^2$ score. On the basis of the results in Section 3, we selected new features that enabled peak-load forecast for these facilities, and propose several methods to improve predictive performance in Section 4. The details and results of our experiments to verify our proposals are described in Section 5. Results showed dramatic performance improvement compared to that in an unpredictable situation. Section 6 summarizes our research and discuss future research directions.

## 2. Related Works

Time-series analysis is a traditional method of designing a mathematical prediction model with historical information. Historical data can either be single variable or multivariable. When forecasting with only single-variable information, most use past peak-load information. When designing a model using multivariable information, on the other hand, meteorological information such as temperature, humidity, and precipitation is mainly used in addition to peak-load information. However, as the amount of feature information increases, modeling becomes complicated, and it is difficult to apply the prediction model implemented on the basis of a predefined mathematical assumption to various environments. Because of the advantage of being able to build an appropriate model without assuming a mathematical model, AI techniques were used rather than time-series analysis. However, time-series analysis is suitable for simplified design and implementation for a specific situation.

Amjady et al. proposed a modified autoregressive integrated moving average (ARIMA) model for short-term load prediction [1]. This approach can accurately predict hourly loads on weekdays, weekends, and holidays, and provides more accurate results than those of existing technologies, such as artificial neural networks or Box–Jenkins models.

The need to maximize predictive performance, collect a variety of information, and develop hardware and software has begun to focus on AI-based prediction methods such as machine learning or deep learning. Time-series analysis uses a fixed mathematical model, so it is difficult to respond to trends or changes in data that change over time. However, an AI technique can be relatively free from the aforementioned problems because it can continuously learn using measurement data. It is also

possible to design a predictive model without background knowledge. For this reason, recent research trends in most fields have been focused on improving prediction performance by using AI algorithms rather than time-series analysis.

D. L. Marino [2] showed that energy management is very important, and if we want to provide or ensure a continuous energy load, we need to focus on managing it. This can be achieved through managing energy and by reducing energy wastage. Therefore, load forecasting should provide an exceptional margin level in managing energy. That paper focused on a deep learning-based method for load forecasting, as the authors presented novel energy load forecasting methodology based on deep neural networks, specifically long short-term memory (LSTM) algorithms. There are two LSTM variants—standard and LSTM-based sequence-to-sequence (S2S) architectures–which were surveyed and implemented on one-hour and one-min time-step resolution datasets. Results showed that LSTM failed at the one-min resolution data, but performed well in the one-hour resolution data, while the performance of the S2S architecture was better. Further, the presented methods produced comparable results with other deep learning methods for energy forecasting in the literature. Additionally, the S2S architecture provides a flexible model to estimate a load. Ouyang T. [3] proposed a deep learning framework to forecast short-term grid load. The framework includes load data that are processed by Box–Cox transformation, and two parameters (electricity price and temperature) were considered. Researchers used parametric Gumbel–Hougaard Copula models, and found the threshold of the peak load to check the dependencies of the power load on the above-mentioned parameters. Then, to forecast the hourly load of the power grid, a deep belief network was built. They independently examined short-term load forecasting in four seasons. They compared their framework with classical neural networks, support vector regression machines, extreme learning machines, and classical deep belief networks. They calculated their load forecast results by mean absolute percentage error, root-mean-square error, and hit rate. Computational results confirmed the effectiveness of the proposed data-driven deep learning framework. The proposed framework could be used for practical short-term scheduling and operations for a grid network. Shi et al. [4] proposed a deep learning framework for predicting household load. As household load prediction is very difficult to predict due to high volatility and uncertainty in the load profile compared to system-level prediction, there are many approaches to remove uncertainty through aggregation, classification, or a spectrum. In this case, overfitting was avoided by applying a pooling-based approach. W. Kong [5] focused on residential load forecasting, as it plays a vital role in smart grids. Several factors, including resident activities or behaviors, temperature, special events, or weekdays, affect forecasting, and continuous changes in such factors make it difficult to accurately forecast an electric load. They also focused on appliance consumption, basically their sequences, and proposed an LSTM-based deep learning forecasting framework. Their work further showed that appliance measurements can improve forecasting accuracy if these measurements are added into the training data. The proposed algorithm creates a significant relationship between consumptions over time intervals. Furthermore, the availability of consumption sequences, specifically of heavy loads, helps in improving meter-level forecasting accuracy under a proposed LSTM framework. S. Motepe [6] proposed a novel hybrid deep learning method for a South African distribution network load forecasting system. The system comprised modules that handled the collection of the loading data from the field, analysis of data integrity by using fuzzy logic, data preprocessing, consolidation of loading and temperature data, and load forecasting. Load forecasting results were then used to notify maintenance planning.

Some studies evaluated special topics such as feature selection, feature extraction, similarity determination, and clustering that indirectly improve both time-series analysis and AI techniques, instead of proposing new algorithms through model modification or combination. The results of these studies can be applied to most algorithms to improve performance.

T. Haida [7] proposed an algorithm for forecasting daily peak loads for a complete year, including holidays, but it was shown that the holiday load was less than workday loads, and this algorithm was based on a regression technique. The authors suggested that, for annual load forecasting, seasonal

factors should be considered during implementation, as they affect forecasting quite a lot. In the winter and summer seasons, an explanatory variable is accordingly added. During spring and fall, they used a transformation technique that dealt with the nonlinear relationship between temperature and load. Last, for better results and an accurate forecast, a holiday-adjustment factor was deducted from the normal load. E. E. El-Attar focused on the problem that many academic institutions spend a lot of money on: power consumption [8]. Therefore, predicting when these peak loads occur to have more uniform energy consumption can help in making more precise strategies. This research was conducted for the Rochester Institute of Technology (RIT), where demand charges are based on a 15-min sliding window per month. This research was conducted for a year, from May 2015 to April 2016, and observed a total of 57 peak days. During this period, the proposed model predicted a total of 74 peak days, 40 of which cases were true positives, and achieved an accuracy level of 70%. Results showed that this forecasting strategy could save up to USD 80,000 for a single RIT submeter. In addition to the hourly load, maximal daily load is an important problem in decentralizing the center of the power grid. Yu et al. [9] showed the prediction method that implemented a gated recurrent neural network (GRNN) by determining similarity using dynamic time warping (DTW) for the load information provided by the load prediction competition organized by the European Network on Intelligent Technologies (EUNITE). D. Jiandong [10] proposed a new combined prediction algorithm to avoid the problem of when the host of a single model cannot meet the demands of high-precision forecasting, as the daily energy consumption sequence is varied by time and random factors. They disintegrated data into intrinsic mode function (IMF) with bandwidth by a variational mode decomposition (VMD) algorithm and extracted feature information. The authors also targeted the speed of prediction and focused on reducing the workload. They tested data with the proposed algorithm, compared them to those of other algorithms, and showed that the proposed algorithm based on VMD-SE could achieve higher prediction accuracy. Cai et al. confirmed the applicability of recurrent neural networks (RNNs) and convolutional neural networks (CNNs) to fill the knowledge gap of deep learning-based technology for predicting the daily load of commercial buildings [11]. Compared to the SARIMAX model, prediction accuracy was improved by 22.6%. In this paper, data on humidity, wind speed, load, external temperature, and air pressure were used in three commercial buildings composed of academic buildings, elementary/secondary schools, and grocery stores at one-hour intervals over the course of one year. As a result of data analysis, uncertainty decreased as building-load size increased, and an unpredictable irregular pattern occurred even at a certain daily pattern. For the correlation between feature values and load, outdoor temperature was selected as the main parameter, and hyperparameters were adjusted for each model by using a training and verification dataset. As a result, both the basic RNN and the CNN [12] produced promising results compared to SARIMAX, especially when the load pattern of the building was very uncertain. Kong et al. proposed an LSTM neural network-based framework for electrical load data for thousands of households through a smart grid smart city project initiated by the Australian government over the course of four years [13]. The data used in this study performed household-level load predictions for 69 customers with hot water systems. Unlike the system level, this lacked a consistent pattern, and there was significant pattern difference, especially because it is influenced by tenants' lifestyle and the type of major appliances they own. The consistency of the daily power profile was evaluated by using a density-based clustering technique, and the LSTM network was used to abstract the state of some residents from the pattern of the input profile, maintain the memory of the state, and make predictions on the basis of the learned information.

There are also peak-load forecasting topics related to cybersecurity issues. Small-scale industrial facilities can be particularly vulnerable to cybersecurity, and attacks that compromise data integrity, such as false data injection, have very negative impact on peak-load forecast performance. This can lead to system failure and loss of operating costs because of incorrect predictions. Jian Luo et al. [14] and Yao Zhang et al. [15] evaluated the prediction performance of various prediction algorithms by using machine learning for GEFCom2012 and GEFCom2014 data, respectively. Both papers showed

that the support vector regression (SVR) algorithm was the most robust, but all models failed to provide accurate load forecasts when the data points were severely contaminated. Therefore, future peak-load forecasting studies should consider not only performance but also cybersecurity issues.

## 3. Data-Limit Analysis in Small-Scale Industrial Facilities

This study used load data from two small industrial facilities located in Silicon Valley, California, USA. The loads were measured at 5-min intervals over the course of two years, from 2017 to 2018, for each facility.

Figures 1 and 2 show the resampling results of daily peak load between total load data for Facilities 1 and 2, respectively. We confirmed that both datasets were nonstationary and that statistical characteristics were different through ADF and KPSS tests. This implied that past peak-load information barely helped to predict future peak load. In this case, it is impossible to implement a generalized model for predicting peak loads for datasets with different statistical characteristics by using a single time-series analysis technique.

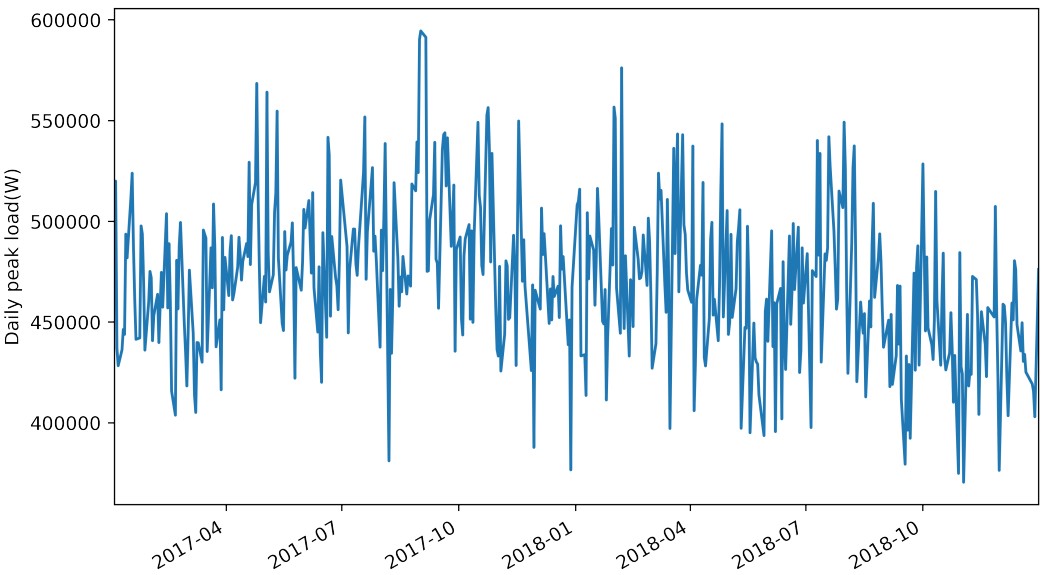

**Figure 1.** Daily peak load over time for Industrial Facility 1 sampled over 2 years.

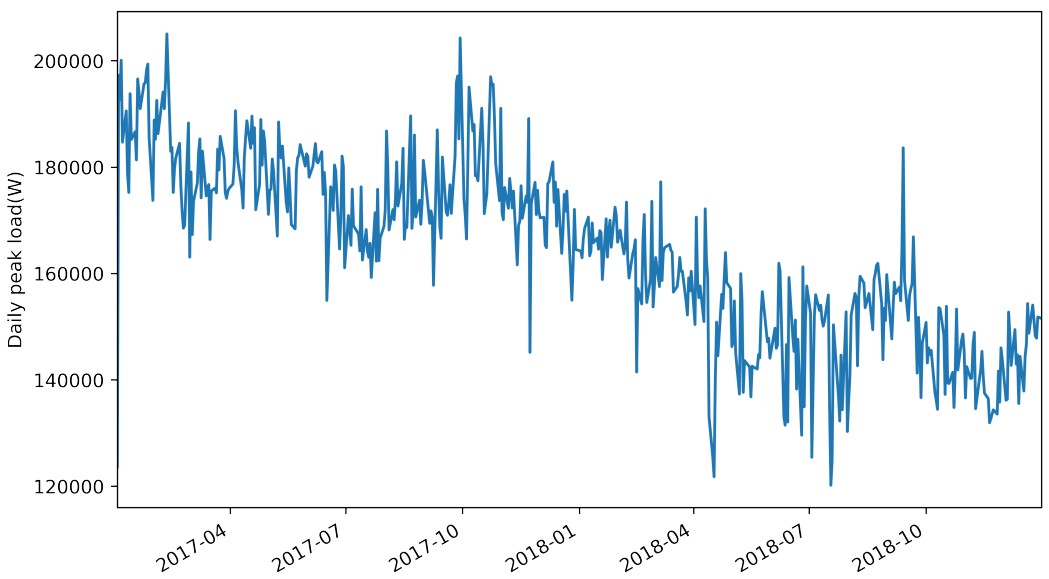

**Figure 2.** Daily peak load over time for Industrial Facility 2 sampled over 2 years.

Because the extra-trees algorithm generally provides near optimal accuracy and good computational complexity, especially on classification and regression problems [16], we applied this algorithm to predict the peak load of our dataset. Figure 3 shows the prediction of future peak load by using the extra-trees algorithm for the two facilities.

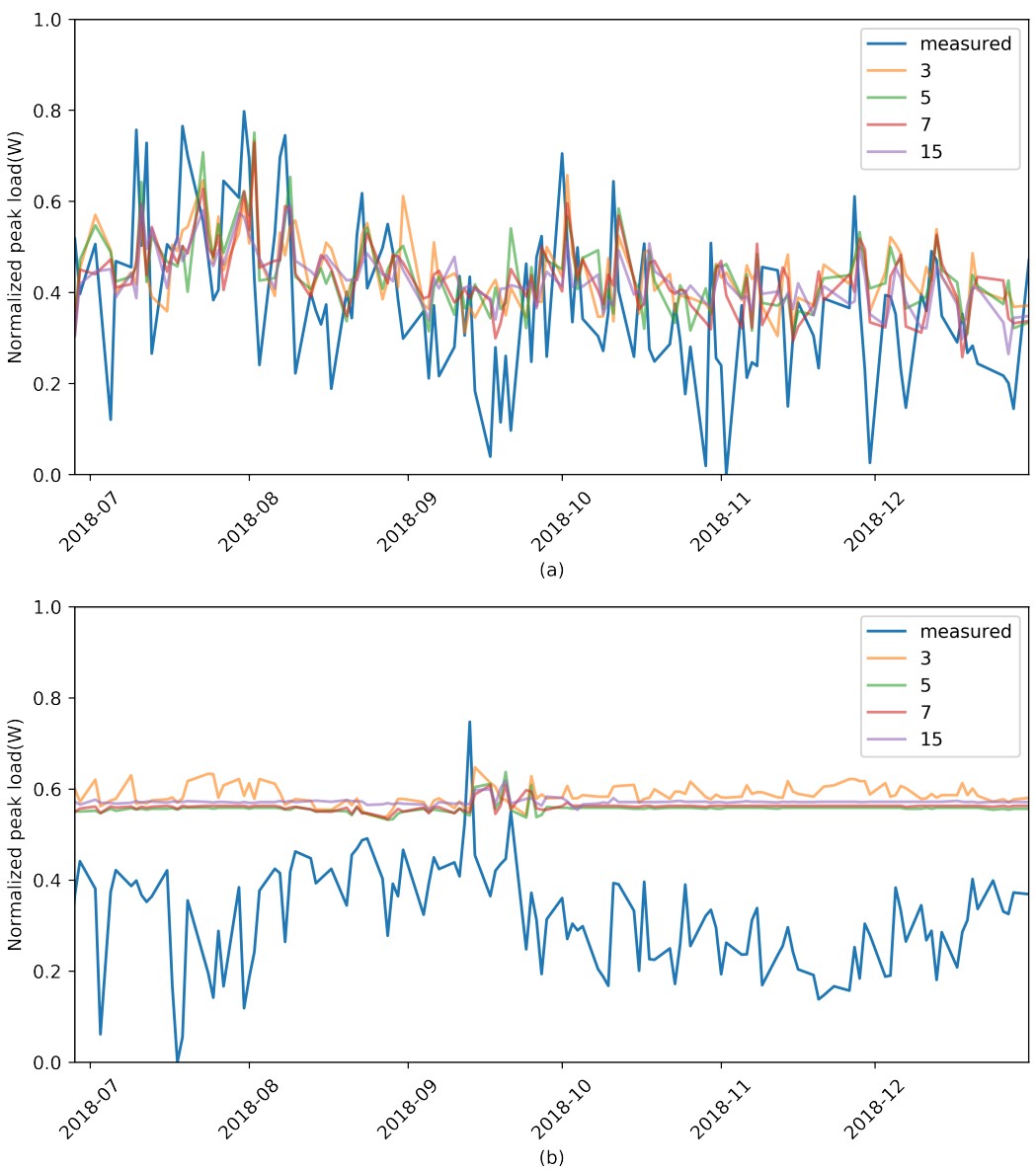

**Figure 3.** Peak-load forecast with extra trees for (**a**) Facility 1 and (**b**) Facility 2 with respect to number of historical data. Increase in number of features did not drastically affect classifier performance.

Within this paper, the $r^2$ score, also called the correlation coefficient; root-mean-square error (RMSE); and mean absolute percentage error (MAPE) were continuously used as metrics to express the forecast performance of various models. As the $r^2$ score is calculated on the basis of the average value of actual observations, when the result of the calculation is a negative value, it means that performance is worse than that of the model predicted by the average value, whereas a positive value means that performance is better than that of the average prediction model. As the $r^2$ score approaches a value of 1, it means that the predicted and actual values are similar. On the other hand, RMSE and MAPE show error values; so, the closer they are to 0, the better performance is. Because RMSE is highly scale-dependent, it is often used to compare the performance of different models on the same data.

Because MAPE uses percentages, it is often used when comparing the performance of models for datasets with different scales. However, because the operation of dividing by actual observations is included, if the observation value is 0 or very small, the result value may appear infinite; therefore, careful attention is required when using this metric.

Table 1 shows the numerical values for forecast performance. We experimented with changing the number of used features (historical data) for learning and prediction. Visually and numerically, the peak-load prediction for the two facilities using the extra-trees algorithm was shown to fail. For the $r^2$ score, values close to zero were obtained. In other words, performance was lower than that of the model on the basis of the average value of the training data. MAPE also showed a very high value, and, especially in the case of Facility 2, a prediction was not done at all. Even if the number of historical data used for prediction change, this does not affect the prediction results. This result allows for us to make two assumptions: historical peak-load information is independent of time, so this information is not useful for prediction, and the model used for load prediction is not appropriate.

**Table 1.** Numerical forecast results of extra-trees algorithm for two facilities' data with historical features. Note: RMSE, root-mean-square error; MAPE, mean absolute percentage error.

| Facility | Number of Historical Data | $r^2$ Score | RMSE | MAPE |
|---|---|---|---|---|
| Facility 1 | 3 | −0.092 | 0.203 | 42.291 |
|  | 5 | −0.059 | 0.199 | 41.572 |
|  | 7 | 0.015 | 0.201 | 42.395 |
|  | 9 | 0.056 | 0.197 | 40.592 |
| Facility 2 | 3 | 0.021 | 0.246 | 63.186 |
|  | 5 | 0.025 | 0.226 | 54.864 |
|  | 7 | 0.180 | 0.229 | 57.697 |
|  | 9 | 0.096 | 0.258 | 63.496 |

In order to be certain about our assumptions, additional tests were conducted by changing the machine-learning models, including with the bagging, random forest, AdaBoost, and gradient-boosting algorithms, while the number of past features was fixed at 5.

Figure 4 and Table 2 show the results of these additional experiments, which were very similar to those of the previous experiment. Thus, we could conclude that it is difficult for past data to act as prediction variables. Therefore, a new approach is needed to predict daily peak load by using only single-variable peak-load information. In the next section, we propose a peak-load prediction algorithm that uses the same day load pattern, not a past peak load, as a feature variable in machine learning.

**Table 2.** Numerical forecast results of extra-trees algorithm for two facility datasets with historical features.

| Facility | Model | $r^2$ Score | RMSE | MAPE |
|---|---|---|---|---|
| Facility 1 | Bagging | 0.022 | 0.199 | 42.469 |
|  | Random forest | 0.009 | 0.201 | 42.558 |
|  | Extra trees | −0.017 | 0.199 | 42.493 |
|  | AdaBoost | 0.0 | 0.202 | 42.924 |
|  | Gradient boosting | −0.073 | 0.200 | 42.816 |
| Facility 2 | Bagging | −1.812 | 0.236 | 57.472 |
|  | Random forest | −1.779 | 0.230 | 58.535 |
|  | Extra trees | −1.872 | 0.240 | 58.574 |
|  | AdaBoost | −2.764 | 0.269 | 69.929 |
|  | Gradient boosting | −1.908 | 0.230 | 56.877 |

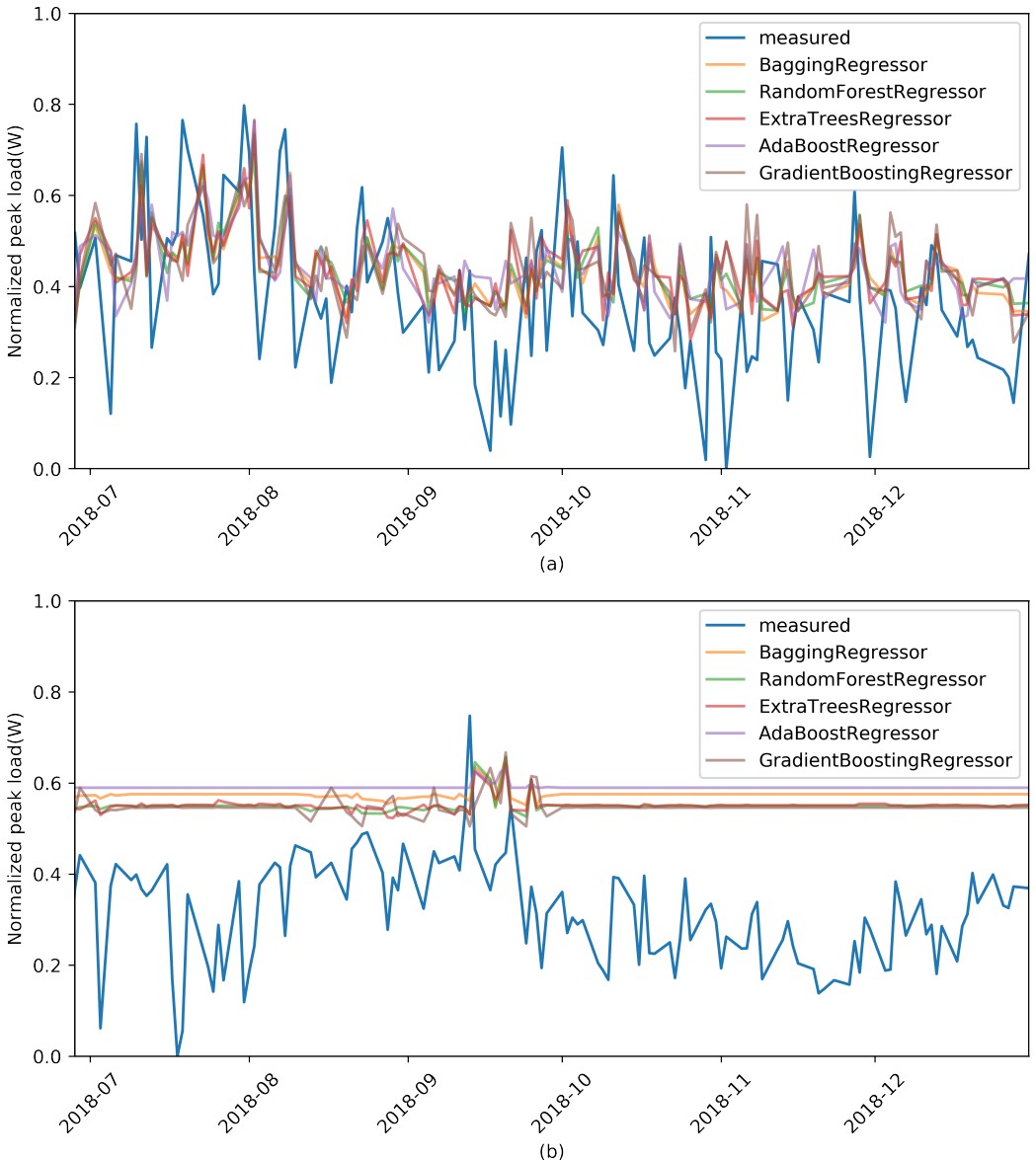

**Figure 4.** Peak-load forecast using different machine learning techniques for (**a**) Facility 1 and (**b**) Facility 2, with number of features set to 5. There was little to no change in classifier performance.

## 4. Proposed Algorithm

In order to make accurate predictions in any application, it is necessary to select an appropriate feature that must be accompanied before implementing a suitable prediction model for general facilities. Features that are not related to prediction have no effect on prediction, and, in some cases, this can significantly reduce performance. Well-known feature information used in a specific field may also not be applicable to other environments. Thus, it is necessary to select new features, and implement a generalized model to apply to small industrial facilities. Therefore, in this section, we study the process of selecting features that are suitable for our data, and propose an algorithm that improves prediction performance compared to that of traditional techniques.

Figure 5 represents our proposed process for predicting the peak load of small-scale industrial facilities that have only power usage information over time. First, we selected new features on the basis of the load pattern given every 5 min rather than daily peak-load information. With these features, we generated several ensemble models, each predicting peak loads and estimating the distribution of training errors. At the same time, we implemented a modified isolation-forest model to accommodate

minor predictions. After that, test samples first estimated the peak-load value by the aggregation of several earlier designed ensemble results. Among these results, the second estimate was returned by adding the compensation value created using error distribution to the sample determined as the outlier. This section covers each procedure in detail.

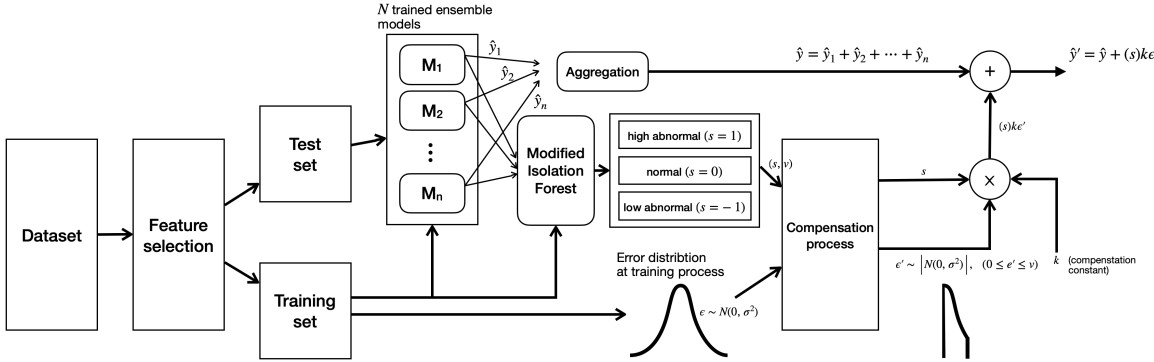

**Figure 5.** Flowchart of the proposed algorithm.

## 4.1. Feature Selection

For most prediction models, including peak-load prediction, an appropriate feature selection and/or extraction process is more important for predicting performance than any other process, including the implementation of prediction models. It is impossible to design a model that can succeed in prediction if the selected feature does not provide information for prediction. Models that use large amounts of feature information require considerable time in the learning and prediction process. In addition, feature information that provides incorrect information degrades prediction performance by disturbing the actual prediction value; thus, it is necessary for it to be able to remove these features in the feature-selection process.

In the previous section, we explained that it is difficult for small-scale industrial facilities to implement a comprehensive forecasting model for a wide range of cases by using historical peak-load information. Therefore, we selected a new feature for predicting peak loads for these facilities on the basis of industrial load assessment data to meet demands across the United States at the Western Interconnection, published in 2013 by the Oak Ridge National Laboratory [17]. The data record the maximal load, hourly electrical energy consumption, and operating hours of approximately 16,000 manufacturing plants in the top 20 industries in the United States. Figure 6 shows the average hourly electrical energy consumption pattern for most manufacturing plants. Most facilities are not run on the weekends, and only a minimal amount of power is uniformly consumed over time. On weekdays, a facility starts operating at 5:00, and high power consumption is maintained between 10:00 and 17:00. Therefore, peak load is very likely to occur between these times. Figure 7 shows the average hourly power usage pattern and the histogram of peak time for the two small-scale industrial facilities that we analyzed. The average power usage pattern on a weekday is slightly different from the Figure 6, but power usage increases from 5:00 and the peak occurring after 10:00 was similar. Thus, power usage patterns can effectively provide peak power information of the same day, although forecast time is later than that of usual peak load forecasting methods that can provide a peak estimate on the previous day. In this case, the question of how many hours of power consumption to select to predict peak load is an important issue. The longer the possible observation time is, the higher the accuracy is, but peak power prediction time becomes too short, which may not be of great help in terms of system operation. This requires an adequate trade-off for performance and peak forecast points.

Histogram information of peak occurrence time gives us a hint about the forecast point. Figure 7 shows that the two industrial facilities had peak power between 10:00 and 22:00, and that there was no

peak power from 05:00 to 08:00. In order to improve performance, we selected power usage data from 05:00 to 09:00 as feature information.

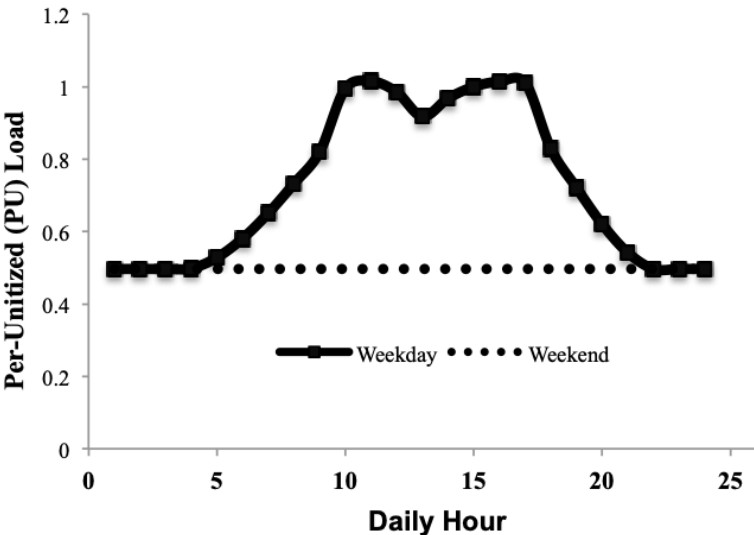

**Figure 6.** Per unit load curve for SIC 32. Source: Assessment of Industrial Load for Demand Response across U.S. Regions of Western Interconnection.

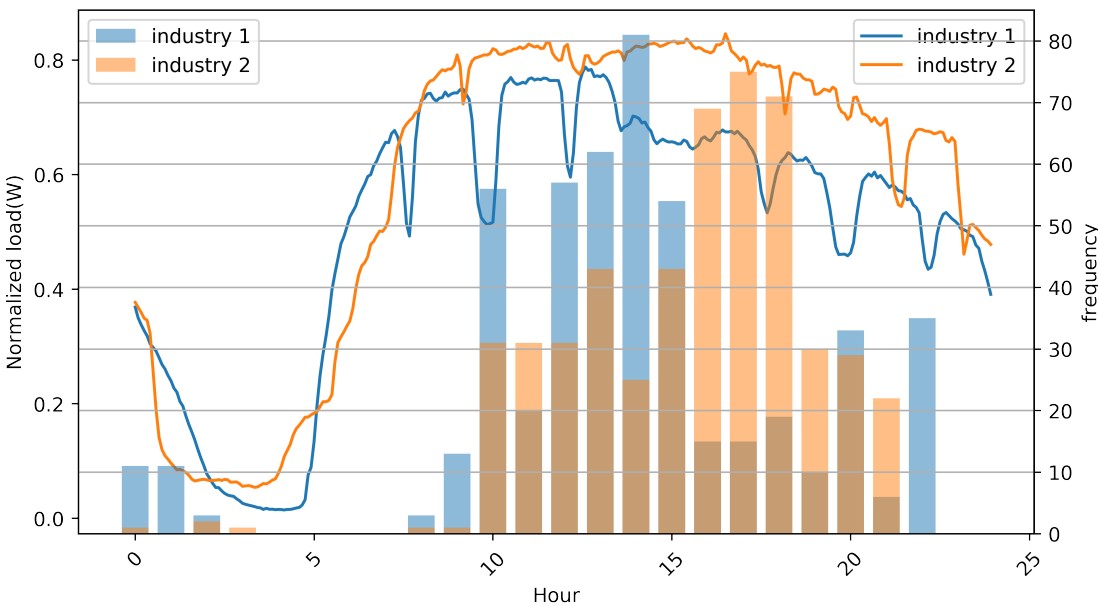

**Figure 7.** Averaged load profile of proposed dataset with average peak-load time between 10:00 and 22:00.

### 4.2. Aggregation of Ensemble Methods

Another important issue for reliable peak-load forecasting is the design of a generalized prediction model that provides accurate forecasts from appropriate feature information. Although various predictive models are being researched and developed to predict peak loads efficiently and accurately, in all cases, there is no single predictive model that performs well. The optimal model should be designed according to data characteristics and the nature of the application. Each industrial facility has its own power usage pattern, and various patterns can coexist in an industrial facility. To cover all of these situations, it is helpful to reconstruct a generalized model through an ensemble of models rather than by using a single prediction model. Ensemble techniques are largely divided into bagging and boosting methods. Both techniques combine multiple decision trees into one powerful predictor

to reduce bias and variance errors. This technique uses the bootstrap sampling method. The difference of the two methods is that bagging is combined in parallel, while boosting is combined in series. Bagging methods such as random forests and extra trees help when we face low variance or overfitting in the model [18]. During the sampling of training data, there are many observations that overlap. Therefore, the combination of these learners helps in overcoming high variance. Bagging is suitable for generalizing the model because it ignores values with the largest and lowest results, and difference can be large, giving an average result.

On the other hand, boosting techniques take care of the weightage of the higher and lower accuracy samples, and then gives the combined results. Net error is evaluated in each learning steps, and it works well with interactions. A boosting technique helps when we are dealing with bias or underfitting in the data set, and it often ignores overfitting or variance issues in the dataset. This technique is useful when designing a model that is most suitable for specific data.

Therefore, aggregating the forecasts of different models can lead to the implementation of a predictive model that can adequately respond to different situations with high accuracy.

*4.3. Uncertainty Compensation*

Although several prediction models produce similar prediction results, if there is a single predictive model that best describes a particular situation, the results of this particular model may be considerably different compared to those of other predictive models. In this case, the aggregation strategy ignores the results of particular prediction models, similarly to the majority-decision strategy, which can lead to a decrease in prediction performance. Therefore, in order to solve this problem, it is necessary to detect a specific situation and correct the aggregated result. Detection of the special situation mentioned above can be resolved by using the isolation forest method. Isolation forest was originally introduced as a method to detect outliers in a dataset. We used isolation forest to improve performance by detecting models that provide different results among the predicted results of several models, and by compensating for anomaly scores of aggregated results. The isolation forest algorithm randomly selects dimensions and divides the space by arbitrary criteria. In the case of a normal value inside a cluster, a large number of space divisions have to be performed in order to completely isolate and leave only one point in the space, whereas outliers separated from the cluster can be isolated by only a small number of space divisions. At this time, the number of outliers and the number of samples located in the space can be appropriately scored for outliers. Isolation forest, as the name suggests, utilizes the above concept by using multiple decision trees. The decision tree must descend deeply until it is completely isolated. Conversely, in the case of outliers, it is highly likely to be isolated even if only the top of the decision tree is burned. Outliers are normalized in the range of 0 to 1, so in general, greater than 0.5 and closer to 1 can be defined as outliers. As each decision tree forms a model by sampling some rather than all data, it also has relatively robust characteristics, even when outliers are close to normal or when they are clustered. We modified this algorithm to properly compensate for large and small outliers, so that they could be divided into normal, high, and low samples. The value of the outlier is normalized into a value between $-1$ and 1; a negative value means an outlier with a lower than normal value, and a positive value means that the outlier has a high value. By applying this outlier value, samples with higher outliers are compensated with more weight in the predicted value. Conversely, samples with lower outliers can improve accuracy by taking less compensation.

## 5. Experiments and Results

To verify that the our proposed algorithm could successfully lead to peak-load forecasting for small-scale industrial facilities with limited information, in this section we outline a series of experiments that we carried out. First, we conducted the same experiment as in Section 3, but input data were reconstructed with the proposed feature information to verify that these new features were actually more useful than historical data were. Then, the degree of performance improvement through the compensation process was tested.

### 5.1. Feature Selection and Generalization

In the previous section, we analyzed patterns of hourly power consumption in small-scale industrial facilities where peak loads are highly dependent on daily production, and we selected new feature information to effectively predict daily peak load. By applying the newly selected features in the same experiment environment as in Section 3, we evaluated the predicted performance of the peak load on our proposed feature information.

Figure 8a,b shows the results of predicting daily peak load with the load pattern before 21:00 that we selected for Facilities 1 and 2, respectively. Bias and variance were lowered through the ensemble of the algorithms used in the previous experiment, and the results were significantly improved compared to those of the previous experiment. Quantitative results for this are shown in Table 3.

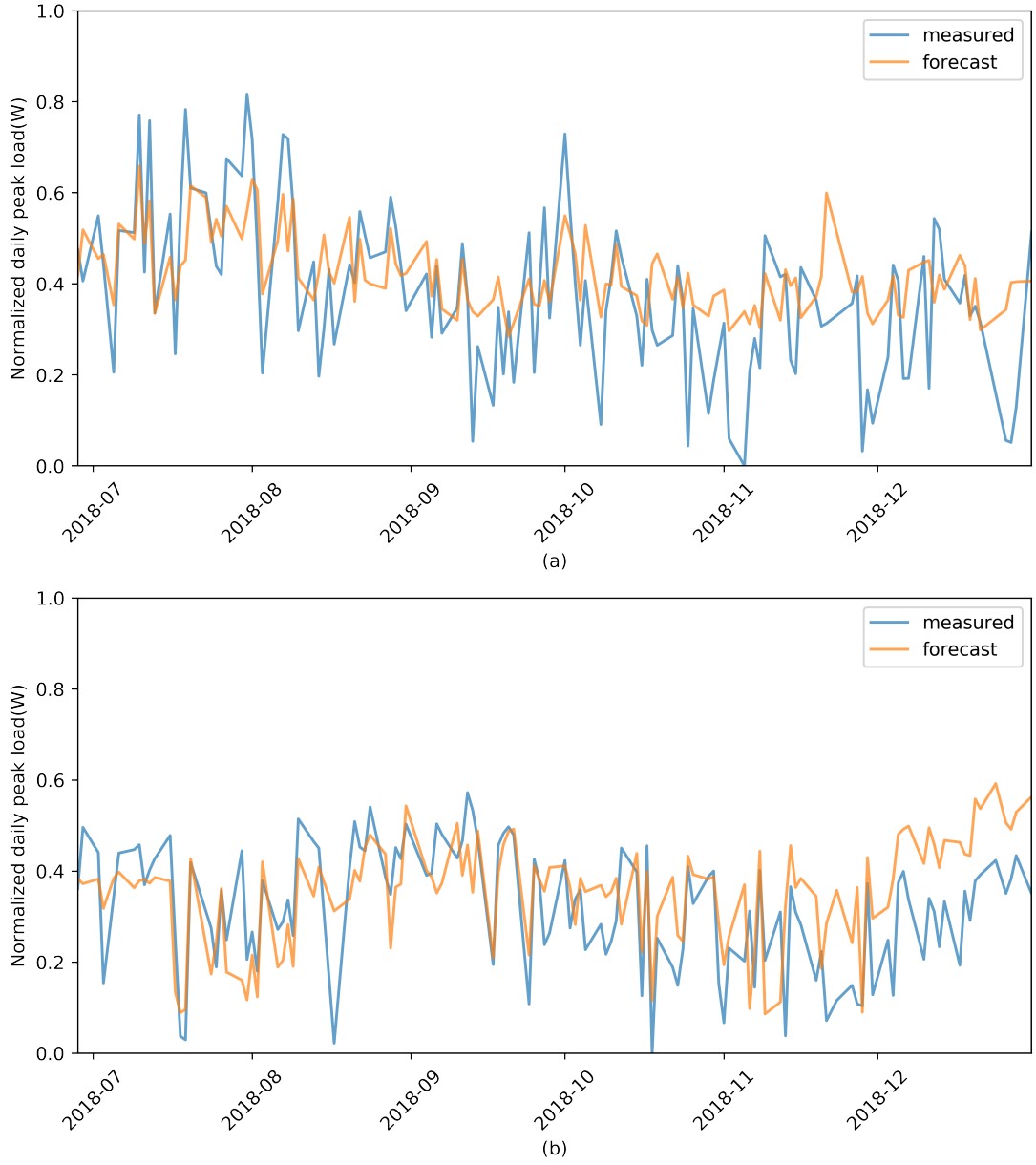

**Figure 8.** Peak-load prediction using proposed ensemble classifier for (**a**) Facility 1 and (**b**) Facility 2. Feature vector for this ensemble was selected as peak load time before 21:00.

**Table 3.** Performance comparison of old and new feature values for prediction.

|  | $r^2$ **Score** |  | **RMSE** |  | **MAPE** |  |
|---|---|---|---|---|---|---|
| **Facility** | **Old** | **New** | **Old** | **New** | **Old** | **New** |
| Facility 1 | 0.016 | 0.306 | 0.167 | 0.216 | 42.816 | 53.077 |
| Facility 2 | −2.002 | 0.463 | 0.231 | 0.173 | 56.877 | 44.048 |

Table 3 quantitatively shows the results of predicting daily peak load on the basis of load data from 05:00 to 09:00. Five prediction ensemble models were trained; then, we aggregated the forecast results by using data from January 2017 to June 2018, and daily peak loads were predicted from July 2018 to December 2018. Intuitively, on days with high production, there was a clear causal relationship between peak load and feature information because power consumption increased rapidly from the time that the facility was operated. However, it is difficult to say that this prediction model was successful in terms of prediction accuracy. On days of high production, the causal relationship between peak power and feature vectors was apparent because power consumption increased rapidly from when the facility was generally operated. Therefore, using these features could help improve peak-load prediction performance. In the case of Facility 2, overall performance improvement occurred, whereas in Facility 1, RMSE and MAPE values were reduced except for the $r^2$ score. We identified this situation as implementing a more general model, and we anticipated that adding a specific process to this generalized model would result in improved performance.

*5.2. Compensation Process*

When the predicted values for each ensemble model are similar, it is likely to be clustered and considered as a normal sample, whereas predicted values that are not clustered and distant for each model are likely to be outliers.

Figures 9 and 10 show the predicted distribution of various models for training data. Predicted distribution was expressed in linear shape and clustered around the mean value. Very large and small values among the results predicted by each model were not clustered; so, when an isolation forest was applied, a high outlier could be detected.

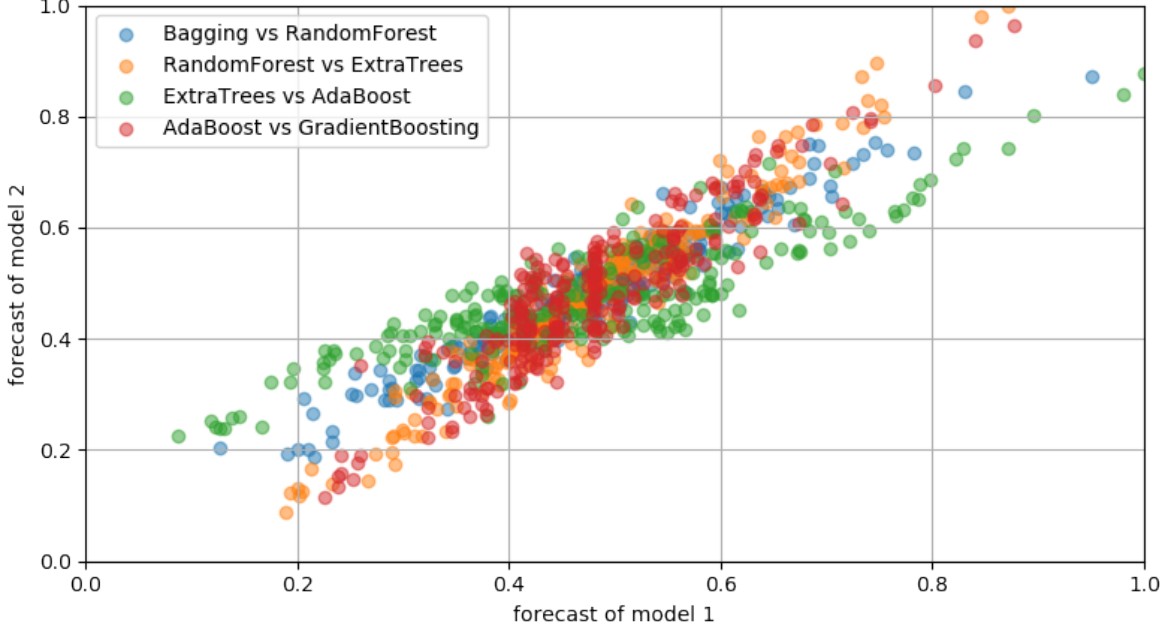

**Figure 9.** Scatterplot of training dataset with different regression models. Overlapped data considered as normal sample around mean, whereas outliers can be seen at plot edges.

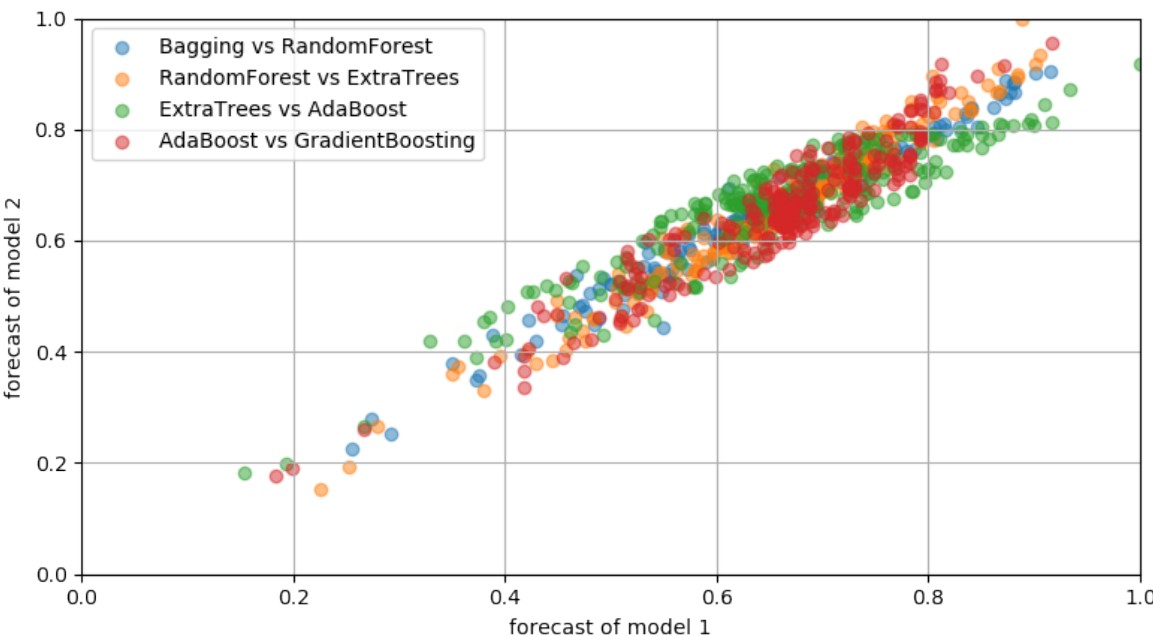

**Figure 10.** Scatterplot of testing dataset with different regression models. Overlapped data considered as normal sample around mean, whereas outliers can be seen at plot edges.

Figure 11 confirms that the prediction result for the test dataset was relatively poor compared to that of the training dataset; particularly, it could not track high and low points well. However, as shown in Figure 10, the distribution of the prediction results for the test set was much more free, so the anomaly score of the isolation forest could be very high, and proper compensation could be provided.

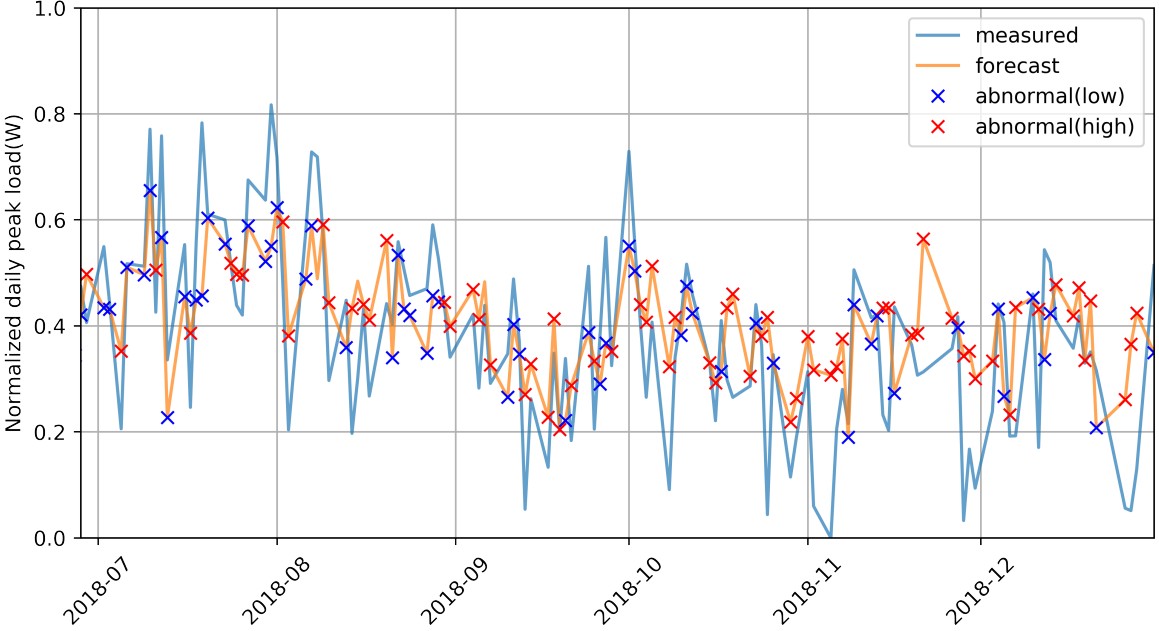

**Figure 11.** Peak-load prediction using isolation forest on test dataset. Network failed to compensate for low and high anomalies.

However, a problem may occur when the predicted result value is larger than the actual value. In this case, the sample is not clustered, and is located farther than the clustered location, so it has a high outlier value. Therefore, as compensation is very large, the prediction result may generate a

larger error than the actual value. In order to prevent this, it is necessary to select an appropriate degree of contamination. If we isolate the abnormal sample from the normal sample using the isolation forest and determine the anomaly score for each sample, we next need to compensate for the predicted results. Initially, in the learning process, random noise is generated using the difference distribution between predicted and actual values. At this time, appropriate range is selected by using the outlier. For instance, if the anomaly score is 0.5, random noise is generated only within the range where distribution is 0.5, and a larger value is not generated. After multiplying the appropriate noise by the appropriate compensation-index value, the compensation value is determined. If the sample has high abnormality, the compensation value is added, and if it has low abnormality, the compensation process is terminated by subtracting the compensation value.

Figure 12 represents error distribution during the training process. The anomaly score of the prediction result is used to determine the random compensation value generated using the distribution obtained in the training process. Different error distributions can be obtained for different datasets. When the outlier score is low, the upper limit of the value that can be generated in the error distribution is lowered. In contrast, when the outlier score is high, the upper limit of this value is increased. In other words, when error distribution can be expressed by $e \sim N(0, \sigma^2)$, and the abnormal score is $v$, this process generates random compensation value $e'$ by using the truncated error distribution between 0 and $v$. Then, if a specific model determines that the prediction result has low abnormality, the generated compression value is subtracted from the aggregation values of the prediction results of various models. Conversely, in the case of high abnormality, the result similar to the actual peak load value can be obtained by adding a compensation value.

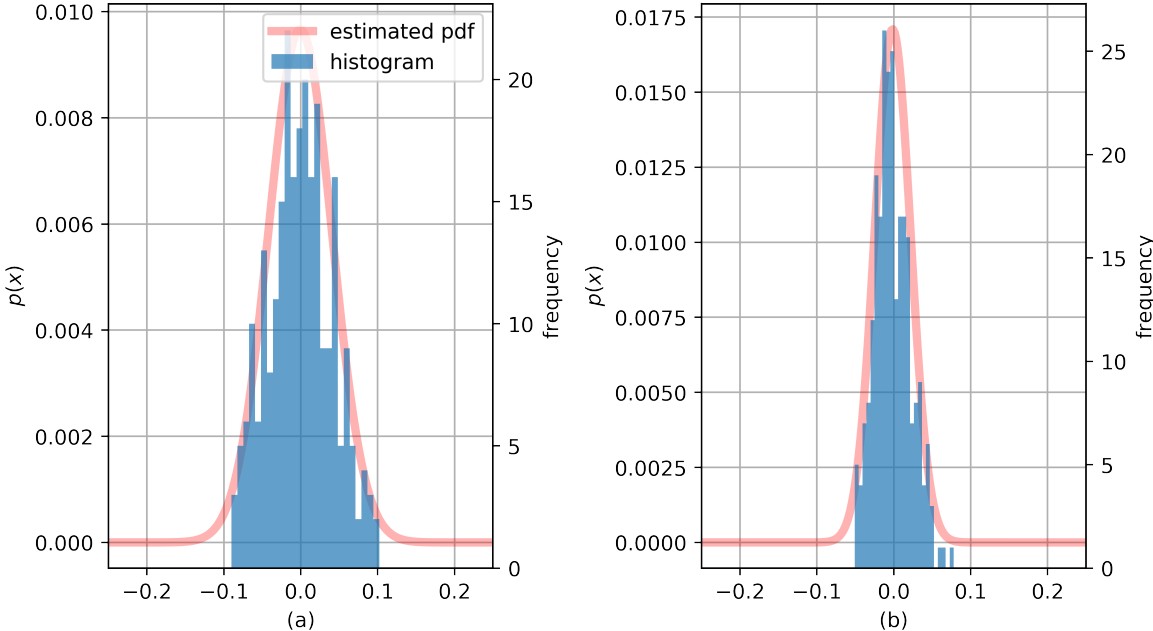

**Figure 12.** Error distribution on training set for (**a**) Facility 1 and (**b**) Facility 2. Probability density function of Facility 1 was more spread compared to that of Facility 2.

It is important to find an appropriate compensation constant for optimal performance. Figure 13 shows the result in terms of accuracy over compensation constant while generating some random noise. In order to find the most appropriate compensation-constant value, we changed the value from 0 to 15 in the learning process, and selected the value with the lowest training error. In this experiment, the value of 4 was obtained as the most effective compensation-constant value for both datasets.

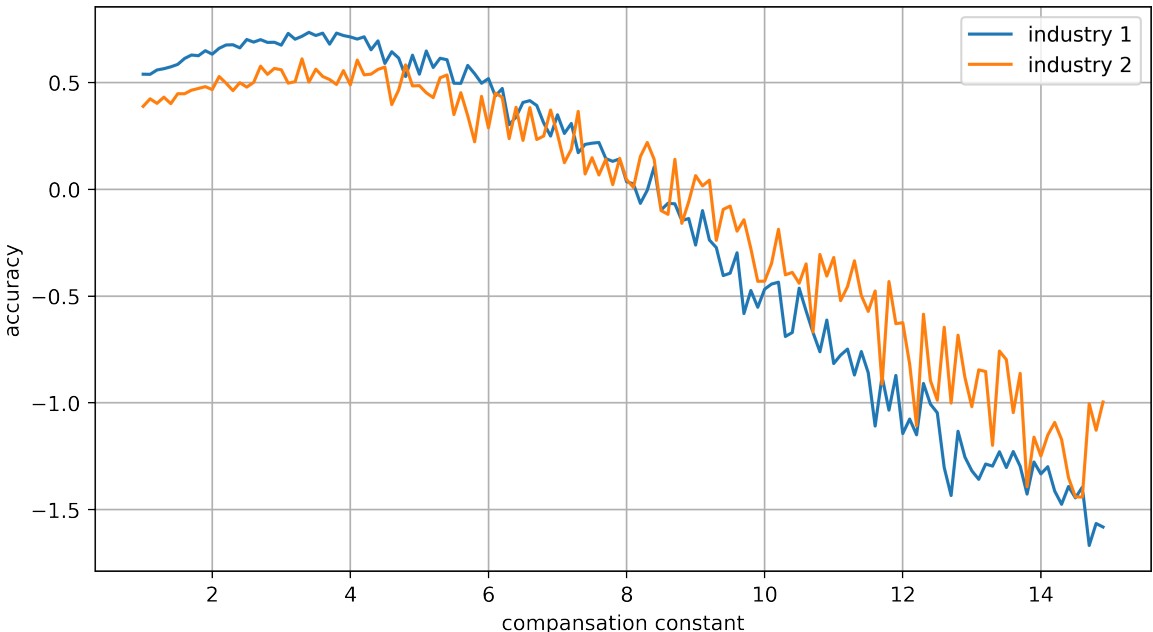

**Figure 13.** Accuracy variation over different compensation constants. Compensation constant value of 4 provided the best performance for our classifier.

The test process was applied using the corresponding compensation constant, and results that compensated for the aggregated prediction values are shown in Figure 14. Compared to Figure 11, both facilities showed similarity between predicted and actual peak values. As mentioned earlier, the two facilities have different statistical characteristics, so when we predicted these datasets through time-series analysis, we needed to implement each mathematical model, but the proposed algorithm seemed to make stable predictions in both cases. Thus, this implies that it could be generalized to various small industrial facilities. As the predicted peak value was larger than the actual peak value, it was also possible to establish in advance an energy-supply plan that could prevent outage due to underestimation.

Table 4 represents the quantitative results for this experiment. It shows that prediction performance numerically increased compared to that in the experiment using the new feature information that we proposed. The prediction results for the two facilities did not show the near-perfect prediction performance for which the recent studies aimed, but they showed significant performance improvement compared to the situation where peak-load prediction was almost impossible due to limited information. Overall, due to the overestimation margin, the number of underestimations was reduced, but there were occasional cases where predictions were still significantly lower than the actual peak load. Further research is needed to improve prediction accuracy.

**Table 4.** Performance comparison of old and new feature prediction values.

| Method | Facility | $r^2$ Score | RMSE | MAPE | Underestimation Number |
|---|---|---|---|---|---|
| Aggregated | Facility 1 | 0.306 | 0.216 | 53.077 | 120 |
| | Facility 2 | 0.463 | 0.173 | 44.048 | 80 |
| Compensated | Facility 1 | 0.732 | 0.095 | 23.486 | 48 |
| | Facility 2 | 0.565 | 0.091 | 23.239 | 50 |

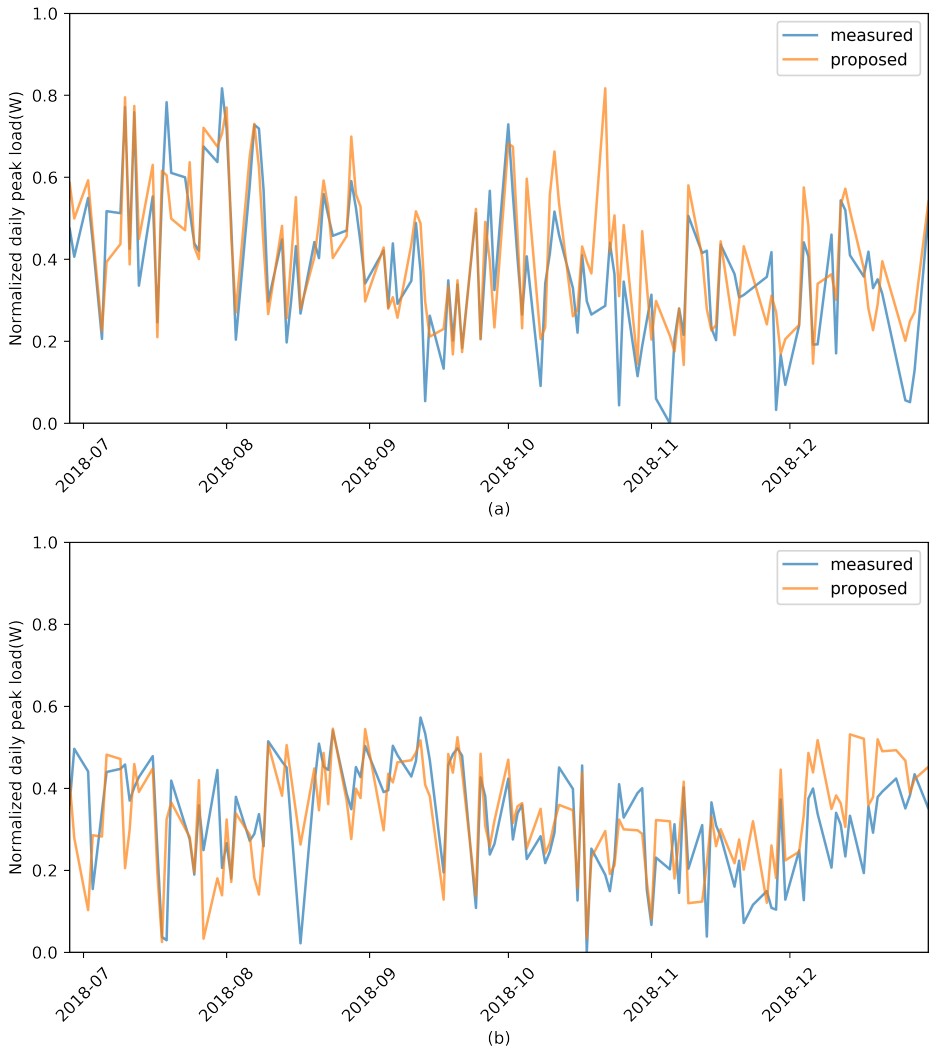

**Figure 14.** Forecasting results of applying all proposed algorithms for (**a**) Facility 1 and (**b**) Facility 2. Compared to Figure 8, the difference between predicted and actual values was significantly reduced, and the underestimation case was also reduced.

## 6. Conclusions

Load prediction is very important in terms of maintaining the system and avoiding problems. This study proposed a machine learning-based algorithm for peak-load forecasting for small-scale industrial facilities that exist only in univariate time-series data with very high uncertainty. The proposed method focused on improving prediction performance by designing a generalized process and adding appropriate compensation to the prediction results of several ensemble algorithms. Experiment results showed that predicted performance significantly increased compared to that of simple machine learning methods. However, forecast performance was still low enough to be applied to practical environments. Aside from experiment results, the prediction method using error distribution obtained in the training process for the aggregated value of disagreement among the predicted values of various models is worth continuously studying. Therefore, we next intend to study how to select the value of the compensation constant used in the experiment in a formal way that can be intuitively understood, instead of a very heuristic method. In addition, we will expand the research to improve actual prediction performance by modifying the compensation process using limited noise. Moreover, artificial neural networks (ANNs), support vector machines (SVMs), and other deep learning-based prediction algorithms will be implemented to study how to robustly respond to cybersecurity problems.

**Author Contributions:** D.-H.K. contributed to methodology, formal analysis, software, data curation, and visualization. N.B.S.Q. contributed to investigation and validation. E.-K.L. contributed to conceptualization, project administration, and validation. All authors contributed to the writing. All authors have read and agreed to the published version of the manuscript.

**Funding:** This work was supported by the National Research Foundation of Korea (NRF) grant funded by the Korea government (Ministry of Science and ICT) (No. 2019R1H1A1101198).

**Conflicts of Interest:** The authors declare no conflict of interest.

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
