# Peer review of "Peak-Load Forecasting for Small Industries: A Machine Learning Approach"

_sustainability, doi:10.3390/su12166539_

Round 1
Reviewer 1 Report
A series of studies have been done in this paper to implement a generalized prediction model applicable to small industrial facilities. The authors clearly define the purpose of the work and highlighted the background of their study. The paper presents many interesting results, which were properly summarized. However, the next version should fix the following technical issues.
1. All figures displayed in this paper are not clear enough now. Please provide high-quality and high-resolution figures when submitting the paper.
2. Observations from Figure 3 and Figure 4 can not correctly reflect the forecasting accuracy of several forecasting approaches. The smart choice is to demonstrate the difference of some error measures (such as root mean square error, RMSE) among different forecasting methods. Such demonstration can be done through some tables.
3. Since input features are very critical for any forecasting problems, some important results and findings from feature selection should be clearly clarified in this paper. For example, which features should be included in peak load forecasting?
4. It looks like that ensemble models are aggregated through the simple average. Is there any other aggregating strategy for ensemble models used in this paper? Any comparisons have been done to compare different aggregating strategies?
5. Literature reviews about energy forecasting are not sufficient now in the current paper. For example, energy forecasting is very vulnerable to false data attacks. Some important papers addressing the robustness of wind power forecasting [1-2] should be included in the literature review.
[1] Benchmarking robustness of load forecasting models under data integrity attacks [J]. Int. J. Forecast. 2018, 34, 89–104.
[2] Robustness of Short-term Wind Power Forecasting against False Data Injection Attacks [J]; Energies, vol. 13 (5), Jul. 2020, 3780.
Author Response
- All figures displayed in this paper are not clear enough now. Please provide high-quality and high-resolution figures when submitting the paper.
The resolution of the figures displayed in the paper was increased. - Observations from Figure 3 and Figure 4 can not correctly reflect the forecasting accuracy of several forecasting approaches. The smart choice is to demonstrate the difference of some error measures (such as root mean square error, RMSE) among different forecasting methods. Such demonstration can be done through some tables.
In response to the comments, we added performance evaluation of the overall experimental results including Figures 3 and 4 through MAPE and RMSE, and we added contents in the paper. - Since input features are very critical for any forecasting problems, some important results and findings from feature selection should be clearly clarified in this paper. For example, which features should be included in peak load forecasting?
As mentioned in the introduction, the reliable accuracy of recent peak load predictions is achieved through a combination of custom models and various feature information. In particular, feature information selection/extraction is still an issue that is still being studied in many ways. However, our research targets small-scale industrial facilities that cannot provide this variety of feature information, and consequently, there is no other information besides load information. In this situation, we were inspired by the daily load pattern of "Assessment of Industrial Load for Demand Response across U.S. Regions of the Western Interconnect". We took the method of predicting the peak load of the day by selecting the load values at the time when the peak occurred as input features. These are written in the paper. - It looks like that ensemble models are aggregated through the simple average. Is there any other aggregating strategy for ensemble models used in this paper? Any comparisons have been done to compare different aggregating strategies?
The aggregation strategy used in this paper fits the simple average strategy of ensemble models. Perhaps it is possible to apply an aggregation strategy using weighted averaging. However, in addition to the simple averaged results, we use the isolation forest to select the results of models with disagreement among each sample prediction. In addition, it was thought that it would provide a value similar to the weighted averaging result because it takes a method to compensate for the value. However, as mentioned, we believe that aggregation strategies that provide more accurate performance are worth researching in the future. - Literature reviews about energy forecasting are not sufficient now in the current paper. For example, energy forecasting is very vulnerable to false data attacks. Some important papers addressing the robustness of wind power forecasting [1-2] should be included in the literature review.
We appriciate your comment. It was very interesting topic. We included two papers in the literature review.
...
There are also peak-load-forecasting topics related to cybersecurity issues. Small-scale industrial facilities can be particularly vulnerable to cybersecurity, and attacks that compromise data integrity, such as false-data injection, have very negative impact on peak-load forecast performance. This can lead to system failure and loss of operating costs because of incorrect predictions. Jian Luo et al.~\cite{b21}, and Yao Zhang et al.~\cite{b22} evaluated the prediction performance of various prediction algorithms by using machine learning for GEFCom2012 and GEFCom2014 data, respectively. Both papers showed that the support vector regression (SVR) algorithm was the most robust, but all models failed to provide accurate load forecasts when the data points were severely contaminated. Therefore, future peak-load-forecasting studies should consider not only performance but also cybersecurity issues.
...
Reviewer 2 Report
In my opinion, this paper describes an experimental method of prediction based on AI for non-stationary data series of high interest and well executed.
However, for a better understanding of the text I highlight the following issues:
1.- In line 69, r-score is mentioned, or does it refer to r2-score?
2.- To avoid mistakes, please define in the text the correlation coefficient r2.
3.- On line 89, the first r2-score should be 0.79, or 79?
4.- In section 4.1. a new feature is selected based on the general industrial consumption data. Could you explain what methodology was used to select this new feature, and why not another one?
5.- In line 374 , it is indicated that the Extra Tree model shows the best performance (r2 = 0.305). Wouldn't it be more correct to say that the Aggregated model shows the best performance (r2 = 0.306)?
6.- On line 384, I think it would be better to say "Compensation process".
7.- In section 5.2. the compensation process through the addition of limited noise is very interesting. Is this procedure based on any previous study?
8.- Regarding figure 13, the method used to select the compensation constant is very heuristic. Is there, in any case, another more formal method for its selection?
9.- The text in figure 14 is wrong. It does not correspond to the figures.
10.- The conclusions in paragraph 6 are obvious. In your opinion, what methods could improve the predictions?
Author Response
- In line 69, r-score is mentioned, or does it refer to r2-score?
As you mentioned, the r2-score is correct, and the typos have been fixed. - To avoid mistakes, please define in the text the correlation coefficient r2.
A brief definition and explanation of RMSE and MAPE along with the description of r2 were added to the paper, and the paper was modified by applying this evaluation method.
...
Within this paper, $r^2$ score, also called the correlation coefficient, root-mean-square error (RMSE), and mean absolute percentage error (MAPE) were continuously used as metrics to express the forecast performance of various models. Since the $r^2$ score is calculated on the basis of the average value of actual observations, when the result of the calculation is a negative value, it means that performance is worse than that of the model predicted by the average value, whereas a positive value means that performance is better than that of the average prediction model. As the $r^2$ score approaches a value of 1, it means that the predicted and actual values are similar. On the other hand, RMSE and MAPE show error values; so, the closer they are to 0, the better performance is. Because RMSE is highly scale-dependent, it is often used to compare the performance of different models on the same data. Because MAPE uses percentages, it is often used when comparing the performance of models for datasets with different scales. However, because the operation of dividing by actual observations is included, if the observation value is 0 or very small, the result value may appear infinite; therefore, careful attention is required when using this metric.
... - On line 89, the first r2-score should be 0.79, or 79?
The content was corrected to 0.73 - In section 4.1. a new feature is selected based on the general industrial consumption data. Could you explain what methodology was used to select this new feature, and why not another one?
The only information we had was the load data (univariate) measured at 5 minute intervals. The method of predicting the future peak load based on the past peak load information has failed, so we have to find another approach. The feature variable was added by obtaining weather information, which is external information, but this method also had the same result. We started searching for a variety of articles, and what we discovered was the "Assessment of Industrial Load for Demand Response across U.S. Regions of the Western Interconnect" material presented in the paper. We were able to get inspiration from the daily load profile, and we decided to simply use it to experiment with the results. We did not write this content in the paper because we did not use any methodology to select new feature information. - In line 374 , it is indicated that the Extra Tree model shows the best performance (r2 = 0.305). Wouldn’t it be more correct to say that the Aggregated model shows the best performance (r2 = 0.306)
Indeed, it is correct that the aggregated model came out with good performance. However, we tried to compare the performance between single ensemble models rather than aggregated model. That is, it was tried to prove that there is no single model that can extract optimal performance for all datasets, and it was written to design a model that can be generalized through aggregation of multiple models. However, as a result of introducing additional metrics such as RMSE and MAPE, Tables 1 and 2 were changed, and the contents were deleted in the process of reconstructing the interpretation. - On line 384, I think it would be better to say “Compensation process”
It was modified to “Compensation process” to reflect the comment - In section 5.2. the compensation process through the addition of limited noise is very interesting. Is this procedure based on any previous study?
The differential privacy, which has been actively researched in the security topic in recent years, knows that a method of increasing the security performance instead of deliberately mixing noise and reducing the prediction performance has been widely used. However, we have not been able to confirm whether special research has been conducted on how to compensate by mixing noise in our research. If there is an existing study on this approach, we will expand the direction of the study based on this. We have written this in the conclusion section.
...
Therefore, we next intend to study how to select the value of the compensation constant used in the experiment in a formal way that can be intuitively understood, instead of a very heuristic method. In addition, we will expand the research to improve actual prediction performance by modifying the compensation process using limited noise.
... - Regarding figure 13, the method used to select the compensation constant is very heuristic. Is there, in any case, another more formal method for its selection?
It is really interesting topic. We are open to the possibility of our research. As a result of studies so far, we know that the predicted performance is quite low. In other words, since our paper was a very experimental subject, it has a strong heuristic factor as well as a question. I plan to confirm the research that rewards me in connection with the previous question, find a more formal method than the heuristic method, and continue the research. This has been added to the conclusion section.
...
Therefore, we intend to proceed in the next study how to select the value of the compensation constant used in the experiment in a formal way that can be intuitively understood, instead of a very heuristic method. In addition, future study will be conducted to expand the research to increase the actual prediction performance by modifying the compensation process using limited noise.
... - The text in figure 14 is wrong. It does not correspond to the figures.
Corrected the caption in Figure 14.
...
Forecasting results of applying all proposed algorithms for (\textbf{a}) Facility 1 and (\textbf{b}) Facility 2. Compared to Figure~\ref{fig:new_features}, the difference between predicted and actual values was significantly reduced, and the underestimation case was also reduced.
... - The conclusions in paragraph 6 are obvious. In your opinion, what methods could improve the predictions?
The feedback you gave us has been very helpful for us. If further research is conducted on the above, it is thought that it will be very helpful to improve the prediction performance. Therefore, these contents were actively reflected and added to the conclusion. Thank you.
...
However, forecast performance was still low enough to be applied to practical environments. Aside from experiment results, the prediction method using error distribution obtained in the training process for the aggregated value of disagreement among the predicted values of various models is worth continuously studying. Therefore, we next intend to study how to select the value of the compensation constant used in the experiment in a formal way that can be intuitively understood, instead of a very heuristic method. In addition, we will expand the research to improve actual prediction performance by modifying the compensation process using limited noise. Moreover, artificial neural networks (ANNs), support vector machines (SVMs), and other deep-learning-based prediction algorithms will be implemented to study how to robustly respond to cybersecurity problems.
...
Reviewer 3 Report
This manuscript employed the machine learning methods to predict the peak load for small industrial facilities. In the proposed method, the isolation forest method was modified to improve the prediction accuracy. Finally, the experimental verifications were conducted to prove the effectiveness of the proposed method. Overall, the topic of this research is interesting and the structure of manuscript was well organized. However, a lot of typos affect the quality of this paper. The detailed comments are provided as follows.
- Please illustrate the main innovation of research. Why was the proposed method considered for task of interests? Why not other machine learning methods like ANN and SVM?
- In the Related Works, when the authors introduced the CNN technology, the following reference is suggested to be included for better demonstrate its principle.
https://doi.org/10.1177/1475921718804132
- In this study, different machine learning methods were compared in terms of r-square. Are there any other indices to evaluate the method? Only one index could not comprehensively evaluate the algorithm performance.
- More future research should be added in the conclusion part.
- There are several obvious typos that affects the paper quality. Here, I list a couple of samples.
Line 8: “aggreagation” should be “aggregation”
Line 89: “073” should be “0.73”
Line 102 “suggests” should be “suggested” or “suggest”
Author Response
- Please illustrate the main innovation of research. Why was the proposed method considered for task of interests? Why not other machine learning methods like ANN and SVM?
NNETAR and SVR, which applied ANN and SVM to regression of time series data, were also considered, but recently, more commonly used machine learning algorithms were introduced. Various deep learning cases were also considered, but our research was quite experimental. To optimize this, the learning time of deep learning was very expensive compared to machine learning, so we excluded it. The contents of the comments will be actively reflected in the next study, and added to the conclusion section.
...
Moreover, artificial neural networks (ANNs), support vector machines (SVMs), and other deep-learning-based prediction algorithms will be implemented to study how to robustly respond to cybersecurity problems.
... - In the Related Works, when the authors introduced the CNN technology, the following reference is suggested to be included for better demonstrate its principle.
https://doi.org/10.1177/1475921718804132
It was added to the related research. - In this study, different machine learning methods were compared in terms of r-square. Are there any other indices to evaluate the method? Only one index could not comprehensively evaluate the algorithm performance.
Through RMSE and MAPE, performance evaluation of all experimental results, including Figures 3 and 4, was added, and the contents were written in the paper. - More future research should be added in the conclusion part.
The feedback you gave us has been very helpful for us.
With a few reviews, we were able to determine the limits of our paper and the direction to go. First, in this paper, a simple aggregation method of the results of ensemble models was applied, but a more formal method instead of a heuristic method was used to determine the compensation constant value as well as whether other aggregation methods help improve performance. I will proceed with research on. This has been added to the conclusion section.
...
However, forecast performance was still low enough to be applied to practical environments. Aside from experiment results, the prediction method using error distribution obtained in the training process for the aggregated value of disagreement among the predicted values of various models is worth continuously studying. Therefore, we next intend to study how to select the value of the compensation constant used in the experiment in a formal way that can be intuitively understood, instead of a very heuristic method. In addition, we will expand the research to improve actual prediction performance by modifying the compensation process using limited noise. Moreover, artificial neural networks (ANNs), support vector machines (SVMs), and other deep-learning-based prediction algorithms will be implemented to study how to robustly respond to cybersecurity problems.
... - There are several obvious typos that affects the paper quality. Here, I list a couple of samples.
Line 8: “aggreagation” should be “aggregation”
Line 89: “073” should be “0.73”
Line 102 “suggests” should be “suggested” or “suggest”
we fixed typos.
Round 2
Reviewer 1 Report
No further comments. Thanks for your revision.
Reviewer 2 Report
The included corrections improve the text.
Reviewer 3 Report
The authors have well addressed the reviewer's comments. I do not have further comments.